# The Association Between Myokines, Inflammation, and Nutritional Status in Patients with Multiple Sclerosis

**DOI:** 10.3390/biom15050703

**Published:** 2025-05-12

**Authors:** Natalia Mogiłko, Sylwia Małgorzewicz

**Affiliations:** 1Department of Neurology and Clinical Neuroimmunology, Specialist Hospital in Grudziądz, 86-300 Grudziądz, Poland; 2Department of Clinical Nutrition, Medical University of Gdańsk, 80-211 Gdańsk, Poland

**Keywords:** multiple sclerosis, inflammation, nutritional status, myokines, irisin

## Abstract

Background: Recent studies indicate that in progressive multiple sclerosis (MS)—an inflammatory and degenerative disease of the central nervous system—the biological pathways associated with these effects remain poorly understood. Changes in body weight, whether presenting as overweight or underweight, as well as alterations in adipose and muscle tissue, together with chronic inflammation, may contribute to the disease and influence its course. Objective: This case–control study aimed to measure inflammatory markers and myokine levels (myostatin and irisin), brain-derived neurotrophic factor (BDNF), and IL-6 in the serum of patients with multiple sclerosis and healthy control and assess whether the myokines and cytokines are associated with nutritional status. Methods: The study included 92 MS patients and 75 healthy volunteers. Nutritional status was assessed using the NRS (Nutritional Risk Screening) 2002 and GLIM (Global Leadership Initiative on Malnutrition) criteria. The risks of malnutrition or malnutrition were diagnosed based on ESPEN recommendations. Body composition analysis was performed using the BIA method with the InBody 120 analyzer. Routine laboratory parameters (albumin, lipidogram) were measured. Myostatin, irisin, BDNF, IL-6, and hsCRP were measured using ELISA methods. Statistical analysis was conducted using Statistica 13.0 software. Comparisons between the two groups were conducted using Student’s *t*-test for normally distributed variables and the Mann–Whitney U test for non-normally distributed variables; the differences between groups were calculated using either ANOVA or the Kruskal–Wallis test. Post hoc analysis by the Bonferroni method was applied. Results: In the MS group, high risks of malnutrition (69.0%) and malnutrition (14.0%) were observed. A statistically significant correlation was found between malnutrition (GLIM) and s-albumin (R = 0.2; *p* < 0.05) and hsCRP (R = 0.23; *p* < 0.05). The MS patient group displayed significantly lower levels of irisin, higher levels of hsCRP, and lower s-albumin compared to healthy volunteers. Malnourished patients with MS exhibited significantly lower irisin levels, as well as higher hsCRP in comparison to MS patients who were at risk or well nourished. The levels of myostatin, BDNF, and IL6 did not differ depending on nutritional status. Irisin correlated with hsCRP (R Spearman = −0.5; *p* = 0.01). Conclusions: Our findings highlight the interplay between chronic inflammation, nutritional status, and myokines level in multiple sclerosis.

## 1. Introduction

Multiple sclerosis (MS) is a chronic autoimmune disease in which the immune system attacks the myelin sheaths surrounding nerves in the brain and spinal cord. Myelin is an insulating material that accelerates nerve signal transmission. Damage to this sheath leads to disruptions in communication between the brain and the rest of the body, resulting in various neurological symptoms. The exact cause of MS is unknown, but it is believed to arise from a combination of genetic and environmental factors. MS occurs more frequently in women than in men and is typically diagnosed in young adults [1].

Adipose tissue has regulatory functions through its production of cytokines and adipokines, whose roles in MS are currently being investigated. Changes in body weight—whether presenting as overweight or underweight—disrupt the cytokine/adipokine network, which may influence MS activity. The regulatory activity of adipose tissue is closely linked to muscle tissue, and studies have shown evidence of lipotoxicity within muscle tissue under chronic inflammatory conditions [2].

Figure 1 illustrates the relationship between nutritional status and the duration of MS. Early in the disease course, patients often present with a normal or excessive body weight. As the disease progresses, unintentional weight loss (including muscle mass) may occur, ultimately leading to conditions such as cachexia, sarcopenia, or sarcopenic obesity. For this reason, assessing nutritional status with recommended methods such as GLIM or NRS 2002 (as proposed by ESPEN) is essential, paying particular attention to unintended weight loss [3].

An essential aspect of MS is the presence of inflammation. Authors emphasize the link between the metabolism of adipose and muscle tissues and chronic inflammation. In multiple sclerosis, both CRP and interleukins are crucial for assessing and monitoring inflammation. Measuring CRP levels can be used to evaluate disease activity and treatment response. Similarly, elevated IL-6 levels are associated with inflammatory activity in MS. However, elevated IL-6 levels were found in patients with progressive MS. IL-6 showed positive correlations with the Expanded Disability Status Scale and the Multiple Sclerosis Severity Score [4].

Myokines like myostatin and irisin may serve as potential markers of early muscle damage in MS. Irisin, a relatively recently discovered protein hormone, is produced by muscles during physical activity and is considered a signaling molecule with various biological effects. Irisin may contribute to the conversion of white adipose tissue into brown adipose tissue, potentially benefiting weight control [5]. It also exerts anti-inflammatory effects and influences neurogenesis and neuronal protection [6,7]. Some reports indicate that irisin levels may be reduced in MS patients, possibly due to decreased exercise capacity and the overall inflammatory state [8].

On the other hand, brain-derived neurotrophic factor (BDNF) is among the most important neurotrophins in the brain, playing a key role in neuronal growth, differentiation, and survival. BDNF is a protein encoded by the *BDNF* gene and is vital for synaptic plasticity—the process underlying learning and memory [6]. It promotes the growth and differentiation of new neurons and synapses, particularly in the hippocampus, a critical region for learning and memory, and contributes to neuronal adaptation in response to various experiences. It also provides neuroprotective effects, aiding in the survival of existing neurons and guarding them against degeneration. Furthermore, BDNF is involved in mood regulation and stress responses. Low levels of BDNF are linked to depression and anxiety, whereas physical activity, which raises BDNF levels, can alleviate these symptoms.

Research suggests that in MS, BDNF may have therapeutic potential by protecting nerve cells from damage caused by inflammatory processes [9], supporting the repair of damaged myelin sheaths—a crucial step in improving neurological function in MS—and helping reduce inflammation, the primary mechanism of tissue damage in the disease [10]. Similarly to irisin, physical activity plays a significant role in regulating BDNF levels. Consistent exercise is one of the most effective means of increasing BDNF [11]. Studies by White L. and Castellano V. have demonstrated that regular exercise can elevate BDNF, which is a myokine linked to enhanced muscle regeneration, function, and brain health [12]. This is particularly relevant to neurodegenerative diseases, as BDNF supports neuronal survival and boosts cognitive function. A link between BDNF levels and diet has also been observed: diets high in omega-3 fatty acids, antioxidants, and vitamins appear to increase BDNF production, while chronic stress and insufficient sleep can lower it [13]. Thus, BDNF is a key factor in brain health, with potential benefits for managing neurodegenerative diseases like MS. Maintaining a healthy lifestyle, including regular physical activity and a balanced diet, can help sustain BDNF levels and enhance nervous system function.

A myokine that influences muscle development and may play a role in disease is myostatin. Also known as GDF-8 (Growth Differentiation Factor 8), myostatin is a protein in the transforming growth factor beta (TGF-β) family. It inhibits muscle cell proliferation and differentiation, preventing excessive muscle growth by balancing muscle synthesis and degradation. With age, myostatin levels may increase, contributing to sarcopenia (the age-related loss of muscle mass and function) [14]. Myostatin also has therapeutic applications in muscle-related diseases, such as muscular dystrophy, where inhibiting myostatin may help build muscle mass. Additionally, manipulating myostatin levels may prove beneficial in treating sarcopenia in older individuals [15]. Physical activity is linked to myostatin levels: high-intensity exercise can lower myostatin, thereby promoting muscle growth [16]. Some athletes naturally exhibit lower myostatin levels, which could contribute to exceptional muscle mass and strength [17]. Research further suggests that myostatin may be involved in regulating glucose and fat metabolism [18] and that altering myostatin levels could affect inflammatory states [19]. Ongoing investigations focus on clarifying whether elevated myostatin worsens muscle weakness in MS patients who already experience muscle weakness due to neurological damage [15].

## 2. Materials and Methods

### 2.1. Aim

The case–control study aimed to measure inflammatory markers and myokine levels (myostatin and irisin), brain-derived neurotrophic factor [BDNF], and IL-6 in the serum of patients with multiple sclerosis (MS) and healthy volunteers and assess whether the myokines and cytokines are associated with nutritional status.

### 2.2. Methods

This study included 92 MS patients (mean age 40.4 ± 11.6 years; F 68/M 24), as part of a previously described cohort [20], and a control group of 75 healthy volunteers (mean age 51.2 ± 13.2 years; F 44/M 31). Recruitment was conducted by a neurologist during routine control visits. The exclusion criterion from the study was as follows: lack of consent and the inability to perform body weight and bioimpedance measurement. Participants were asked to review and sign consent forms prior to study initiation.

MS diagnosis was made according to the McDonald criteria [21]. No restrictions were placed on inclusion criteria regarding MS phenotype. The exclusion criterion was the lack of consent to participate in the study or the inability to undergo bioimpedance measurement. Ethical approval was obtained from the MUG Ethical Committee (no. NKBBN/78/2019).

### 2.3. Nutritional Status Assessment

The NRS 2002 (Nutritional Risk Score) includes a screening assessment of the patient’s nutritional status based on BMI, unintentional weight loss, and food intake on a scale of 0 to 3 points, and then also the presence and severity of disease on a scale of 0 to 3 points. Patients receive an additional 1 point for age over 70 years. The GLIM (Global Leadership Initiative on Malnutrition) criteria include the following: Phenotypic criteria: unintentional weight loss—loss of body weight > 5% in the last 6 months or >10% over a period of more than 6 months; low BMI—BMI < 20 kg/m^2^ for people under 70 years of age and <22 kg/m^2^ for people over 70 years of age; And reduced muscle mass—assessed by methods such as bioelectrical impedance analysis (BIA), dual-energy X-ray absorptiometry (DXA), or muscle circumference measurements. Etiologic criteria: reduced food intake or absorption—consuming ≤ 50% of energy requirements for more than 1 week or any reduction in food intake lasting more than 2 weeks; And disease burden or inflammation—presence of acute or chronic inflammation associated with diseases. Patients may receive 1 point for the etiologic criterion and 1 point for the phenotypic criterion.

Body Mass and Body Composition: Measurements of body weight, fat mass, muscle mass, total fat, and total body water were performed using the BIA method with an InBody 120 analyzer (InBody, Warszawa, Poland).

Body Mass Index (BMI): BMI was calculated using the formula BMI = weight (kg)/[height (m)]^2^, along with the Fat-Free Mass Index (FFMI, kg/m^2^).

### 2.4. Nutritional Indices

Malnutrition was diagnosed in patients who scored ≥ 3 points in the NRS 2002 or 2 points in GLIM.

Risk of malnutrition was diagnosed in patients who scored 1 or 2 points in NRS 2002 or 1 point in GLIM.

Fat Free Mass Index (FFMI) < 14 kg/m^2^ was considered reduced.

Obesity was diagnosed based on the World Health Organization (WHO) classification as BMI ≥ 30 kg/m^2^.

A decreased serum albumin value was defined as <3.5 g/dL according to the laboratory recommendation.

### 2.5. Biochemical Parameters

The following biochemical parameters were measured in serum using routine laboratory methods: albumin and lipid profile, ELISA was used to measure myostatin, irisin, and BDNF (kits from Immunodiagnostic, Frankfurt, Germany), IL-6 (Bio-Techne, Warszawa, Poland), and hsCRP (DRG MedTek, Warszawa, Poland).

### 2.6. Statistical Analysis

Results were expressed as mean ± standard deviation or as percentages (for categorical variables). The Shapiro–Wilk test was used to assess normality of distribution. Comparisons between two group were conducted using Student’s *t*-test for normally distributed variables and the Mann–Whitney U test for non-normally distributed variables; the differences between groups were calculated using either ANOVA or the Kruskal–Wallis test. Post hoc analysis by the Bonferroni method was applied. Pearson’s or Spearman’s correlation coefficients were calculated to assess relationships between variables. Independent associations among variables were assessed with stepwise multiple regression analysis. A *p*-value < 0.05 was considered statistically significant. Statistical analyses were performed using STATISTICA PL v 13.0 (StatSoft, Kraków, Poland).

## 3. Results

### 3.1. Comparison of MS Patients and Healthy Controls

According to the GLIM criteria, a substantial proportion of MS patients in the study were at risk of malnutrition (*n* = 63; 69%), while malnutrition was diagnosed in 13 individuals (14%). Meanwhile, 15 patients (16%) were classified as well-nourished NRS 2002, showing a risk of malnutrition in 81% of MS patients and malnutrition in 3.2% (*n* = 3). (Figure 2). Patients received an average of 0.58 ± 0.6 points in NRS 2002 and 1.17 ± 0.37 points in GLIM.

In the studied MS group, 10.8% (*n* = 10) of patients reported unintentional weight loss (in range ≥ 5%), and BMIs < 21.5 were observed in 10.8% (*n* = 10) of patients. The healthy volunteers (control group) demonstrated good nutritional status (0 points in both: NRS2002 and GLIM), with a BMI > 21.5. They did not experience unintentional weight loss, all consumed 100% of their meals, and they did not suffer from diseases that increase the risk of malnutrition.

As shown in Table 1, the MS patient group had significantly lower irisin levels, higher hsCRP levels, and lower s-albumin compared with the control group. Additionally, differences in IL-6 were at the threshold of statistical significance. The BMI, FM, and muscle mass did not differ significantly; however, the FFMI was significantly lower in the MS group.

### 3.2. Comparison of Malnourished MS Patients and Patients at Risk

As shown in Table 2, malnourished patients exhibited unintentional body weight loss, lower BMIs, and significantly lower FM, MM, and FFMI levels in comparison to patients at risk.

Malnourished patients with MS exhibited significantly lower irisin levels and higher hsCRP compared to at-risk and health patients (Table 2). In Figure 3 and Figure 4, respectively, median levels of hsCRP and irisin are presented according to nutritional status assessments based on NRS 2002.

Additionally, a statistically significant correlation was observed between the GLIM results and the s-albumin (R_Spearman = 0.2; *p* < 0.05) and hsCRP (R_Spearman = 0.23; *p* < 0.05) levels. Also, irisin correlated with hsCRP (R_Spearman = −0.5; *p* = 0.01). The levels of myostatin, IL-6, and BDNF did not differ significantly according to nutritional status.

Lipid profile: MS patients exhibited higher LDL cholesterol levels compared to healthy controls, whereas malnourished MS patients displayed lower total cholesterol and LDL cholesterol levels than the other individuals. HDL cholesterol levels did not differ significantly between the study groups.

## 4. Discussion

### 4.1. Nutritional Status and Inflammation

Our study highlights a high risk of malnutrition in patients with multiple sclerosis (MS), accompanied by mild to moderate inflammation. Elevated hsCRP correlated with both s-albumin levels and the GLIM assessment of nutritional status, which may affect appetite and food intake in chronically ill patients. According to the GLIM criteria, disease-related malnutrition was identified, which is known to be associated with poor quality of life, increased mortality risk, a higher incidence of falls, disability, weakened immunity, muscle loss (including respiratory muscles), ventilation disorders, impaired cough reflex, and pneumonia.

Previous research links higher hsCRP levels with an increased risk of malnutrition, worse clinical outcomes, and muscle catabolism [20,22].

### 4.2. Myokines, Body Composition, and Inflammation

In our study, we found a relationship between inflammation and irisin, which was reduced in the MS group as well as in malnourished MS patients in comparison to healthy volunteers. Irisin, a myokine secreted during exercise, plays a key role in muscle function. Studies indicate that physical activity boosts irisin secretion, which in turn improves muscle function, reduces inflammation, and supports cardiovascular health [23]. In a study by Tu W-J et al. on stroke patients, lower irisin levels were associated with worse neurological outcomes, larger infarct volumes, and higher CRP and IL-6 levels, suggesting that irisin may exert a protective effect on both muscles and the brain [24].

Research shows that resistance exercise combined with amino acid supplementation can significantly influence levels of myokines such as irisin and myostatin. These interventions enhance muscle quality, reduce inflammation, and decrease body fat, suggesting a synergistic effect of nutrition and physical activity in regulating myokines and inflammation [25]. Many studies have demonstrated that physical activity can reverse at least some negative effects of a sedentary lifestyle, potentially delaying brain aging and degenerative diseases such as Alzheimer’s disease, diabetes, and multiple sclerosis.

Furthermore, researchers have examined irisin levels in serum and cerebrospinal fluid in MS patients and in EAE (Experimental Autoimmune Encephalomyelitis) mouse models. Serum irisin was elevated in MS patients and EAE mice, whereas FNDC5 (fibronectin type III domain-containing protein 5)/irisin mRNA levels were decreased in both the spinal cord and brain, regardless of whether EAE was at the onset, peak, or chronic phase. Irisin concentrations shifted with disease progression in both MS and EAE. Authors concluded that irisin may be involved in the pathological processes of MS/EAE [7].

In our study, we did not observe differences in BDNF and myostatin levels between MS patients and healthy controls, likely due to relatively small sample sizes. Other authors highlight the neuroprotective role of BDNF, pointing out its ability to reduce pro-inflammatory mediators (e.g., cytokines) and stimulate myelin repair [10]. Kerschensteiner et al. demonstrated the role of BDNF in oligodendrocyte regeneration and myelin repair [26]. Weinstock-Guttman et al. found that BDNF production by immune cells was linked to increased inflammatory activity in white matter and microscopic damage in Normal-Appearing White Matter (NAWM) in MS patients, indicating that BDNF secretion by immune cells may be related to early MS pathology [27]. Patenella et al. reported extended task performance times for divided attention and visual scanning; high IL-6 levels were also linked to lower Mini-Mental State Examination scores in relapsing-remitting MS patients. The effect of both BDNF and IL-6 on cognitive function in MS patients was independent of demographic and clinical characteristics as well as depressive mood disturbances [28].

Recent studies have emphasized the complex link between myostatin and inflammation in various diseases. In rheumatoid arthritis, elevated myostatin levels positively correlate with disease activity and inflammatory markers while negatively correlating with muscle mass [29]. Myostatin induces pro-inflammatory cytokines, such as TNF-α and IL-1β, in rheumatoid arthritis synovial fibroblasts through distinct signaling pathways [30]. In chronic kidney disease, myostatin inhibition suppresses systemic inflammation, reverses muscle wasting, and improves protein metabolism [31]. These findings suggest a multifaceted interaction wherein myostatin acts both as a mediator and target of inflammatory processes, potentially guiding novel therapeutic approaches for conditions marked by chronic inflammation and muscle wasting (Figure 5).

In summary, our results suggest a relationship between irisin, inflammation, and nutritional status in stable MS patients at various stages of the disease. Malnourished patients presented elevated hsCRP and decreased irisin levels. They also presented unintentional body weight loss, decreased appetite, and low BMI with decreased anthropometric indices of body composition. We can assume that the inflammation caused decreased appetite and then caused a loss of body weight, muscle mass and fat mass, and increased risk of malnutrition. On the other hand, chronic inflammation may be an additional factor together with elevated LDL cholesterol, contributing to the increased risk of cardiovascular disease in multiple sclerosis.

In our study, we observed that irisin levels are associated with inflammation, while previous studies indicate that physical exercise increases blood irisin levels and that serum irisin levels are higher in obese individuals. There may be many reasons for reduced serum irisin in individuals with MS, including a lack of physical activity, a decrease in muscle and fat tissue (malnutrition), chronic inflammation, pharmacological treatment, or even an improper diet, but determining these relationships requires further research. Based on the literature and our findings of low irisin levels in MS patients, we suggest that a comprehensive approach—incorporating pharmacotherapy along with tailored physical activity and dietary interventions—may help modulate inflammation and prevent muscle loss.

A limitation of our study is that patients were at varying stages of treatment and disease progression. Additionally, the time since MS diagnosis differed among participants, and measurements were taken only once. These factors may have influenced the results, emphasizing the need for long-term studies.

## 5. Conclusions

Our findings highlight the interplay between chronic inflammation, nutritional status, and myokines level in multiple sclerosis.

## Figures and Tables

**Figure 1 biomolecules-15-00703-f001:**
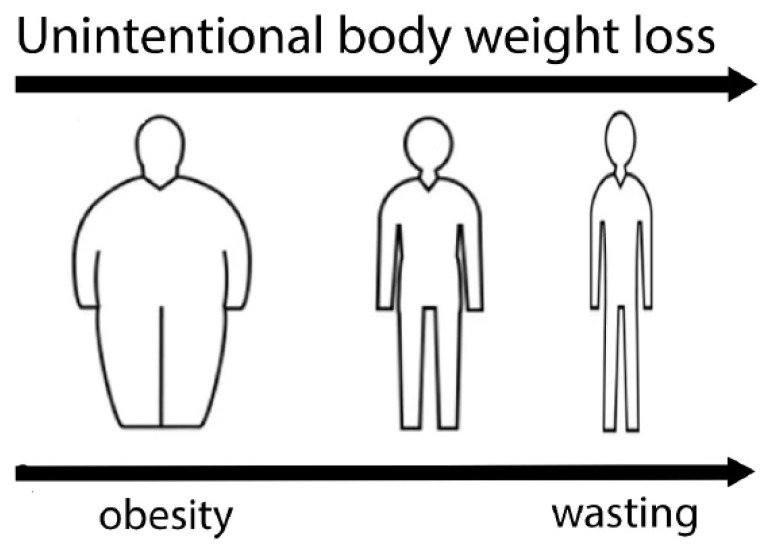
Changes in nutritional status during the course of the disease.

**Figure 2 biomolecules-15-00703-f002:**
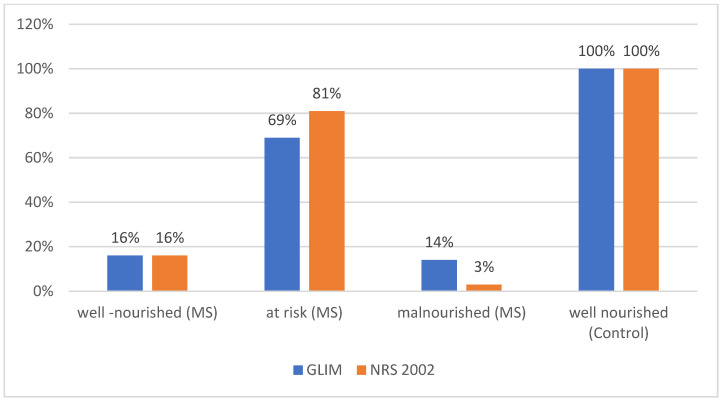
Proportion of patients who were well nourished, at risk of malnutrition, or malnourished in the MS group, and healthy volunteers according to GLIM and NRS 2002 assessments.

**Figure 3 biomolecules-15-00703-f003:**
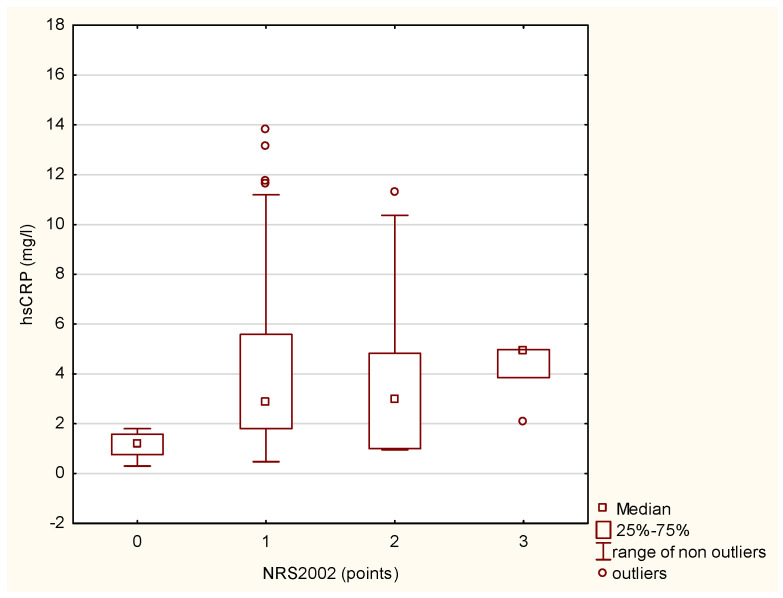
The differences in irisin levels in MS patients according to nutritional status (NRS 2002) using the Kruskal–Wallis test (Chi^2^ = 10.97; df = 3 *p* = 0.01).

**Figure 4 biomolecules-15-00703-f004:**
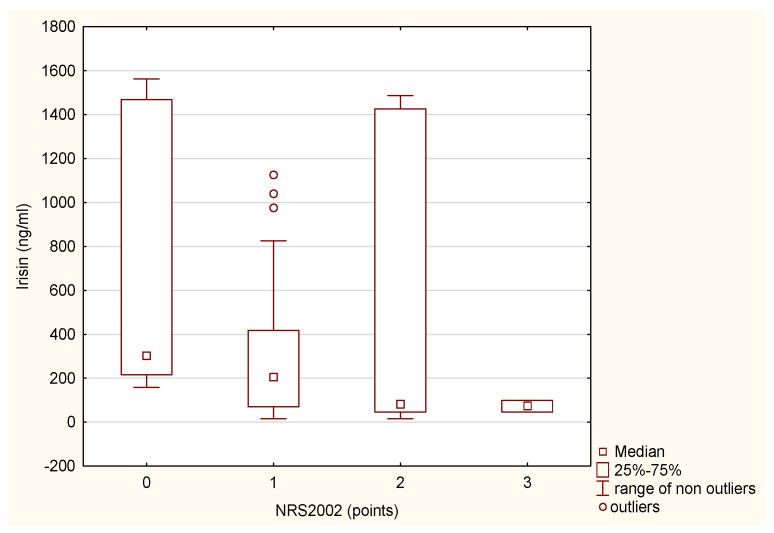
The differences in hsCRP levels in MS patients according to nutritional status (NRS 2002) using the Kruskal–Wallis test (Chi^2^ kwadrat= 11,88; df = 3 *p* =0.00).

**Figure 5 biomolecules-15-00703-f005:**
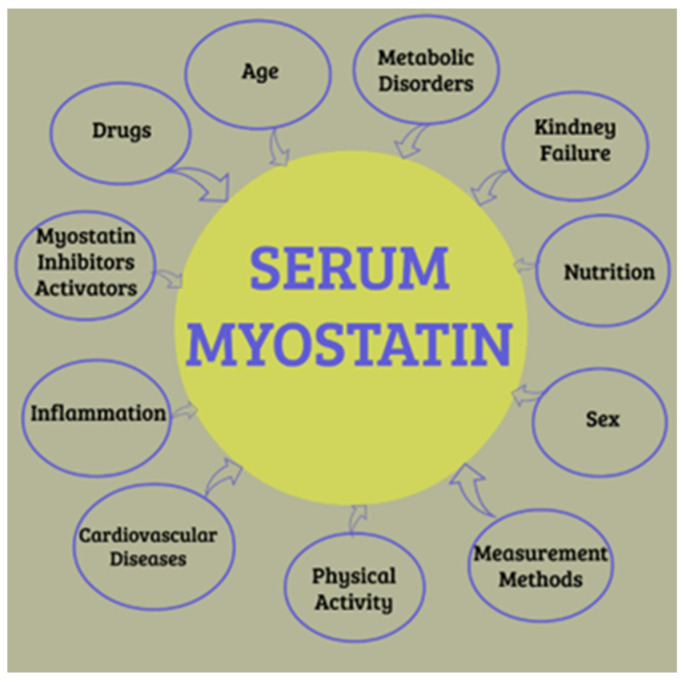
Various factors affecting serum myostatin levels. Adapted from Baczek, et al. 2020 [32].

**Table 1 biomolecules-15-00703-t001:** Comparison of the studied parameters between the MS and control groups.

	(median 1.77)	(median 1.73)	
	Min-Max: 0.4–5.01	Min-Max: 0.4–4.67	

**Parameters**	**MS (** * **n** * **= 92)**	**Healthy Control (** * **n** * **= 75)**	* **p** * **-Value**
Age (yr.)	46.57 ± 11.7	51.2 ± 13.2	*p* = 0.05
	(median:46)	(median:61)	
	Min-Max 24–71	Min-Max 35–70	
Body weight (kg)	74.50 ± 16.73	74.83 ± 14.70	*p* = 0.78
	(median: 73.60)	(median: 73.60)	
	Min-Max: 45.00–130.00	Min-Max: 49.40–120.00	
Fat mass (kg)	23.77 ± 10.17	25.72 ± 9.32	*p* = 0.45
	(median: 24.83)	(median: 24.83)	
	Min-Max: 5.80–65.90	Min-Max: 7.60–44.00	
Muscle mass (kg)	30.20 ± 7.40	28.90 ± 6.90	*p* = 0.35
	(median: 29.70)	(median: 28.90)	
	Min-Max: 16.00–50.00	Min-Max: 15.00–47.00	
Body fat (%)	31.3 ± 8.7	35.4 ± 7.4	*p* = 0.61
	(median: 31.7)	(median: 33.0)	
	Min-Max: 13.2–50.7	Min-Max: 13–50	
Total water (L)	37.1 ± 8.2	35.4 ± 7.4	
	(median: 34.6)	(median: 33.7)	
	Min-Max: 24.4–59.8	Min-Max: 14.5–52.4	
BMI	25.83 ± 5.05	27.35 ± 4.35	*p* = 0.12
	(median: 25.60)	(median: 27.10)	
	Min-Max: 16.70–43.40	Min-Max: 20.30–37.80	
UBWL (kg)	1.75 ± 2.53	0	*p* = 0.00
	(median 1.75)	(median: 0)	
	Min-Max: 3.31–6.81	Min-Max: 0	
Total cholesterol (mg/dl)	200.2 ± 45.6	198.3 ± 42.3	*p* = 0.65
	(median: 201.0)	(median: 198.3)	
	Min-Max: 140.0–300.0	Min-Max: 130.0–290.0	
HDL cholesterol (mg/dl)	61.9 ± 16.4	57.9 ± 14.3	*p* = 0.13
	(median: 60.8)	(median: 68)	
	Min-Max: 37.8–131	Min-Max: 22.8–87	
LDL cholesterol (mg/dl)	130 ± 39.8	69.8 ± 26.2	*p* = 0.04
	(median: 124)	(median: 69.8)	
	Min-Max: 53–242	Min-Max: 40–177	
Albumin (g/L)	39.4 ± 4.0	41.2 ± 3.8	*p* = 0.02
	(median: 40.0)	(median: 41.2)	
	Min-Max: 25.0–47.0	Min-Max: 28.0–45.0	
hsCRP (mg/L)	4.4 ± 13.4	2.1 ± 7.2	*p* = 0.03
	(median: 2.6)	(median: 2.1)	
	Min-Max: 0.5–40.0	Min-Max: 0.2–20.0	
Irisin (ng/mL)	497.21 ± 690.1	765.85 ± 799.35	*p* = 0.02
	(median: 181.54)	(median: 369.39)	
	Min-Max: 15.84–2714.38	Min-Max: 15.9–2704.4	
IL-6 (pg/mL)	1.86 ± 3.12	1.32 ± 1.22	*p* = 0.49
	(median: 0.97)	(median: 1.32)	
	Min-Max: 0.22–21.8	Min-Max: 0.30–3.76	
BDNF (pg/mL)	387.47 ± 647.82	365.77 ± 436.21	*p* = 0.90
	(median: 387.47)	(median: 365.77)	
	Min-Max: 0.40–1683.11	Min-Max: 0.41–1238.19	
Myostatin (ng/mL)	1.77 ± 1.62	1.73 ± 1.47	*p* = 0.93
	(median: 1.77)	(median: 1.73)	
	Min-Max: 0.4–5.01	Min-Max: 0.4–4.67	

Data are expressed as mean ± SD, median, and range. Student-t and Mann–Whitney U tests were used for comparisons between the MS group and healthy volunteers.

**Table 2 biomolecules-15-00703-t002:** Comparison of studied parameters in malnourished vs. at-risk patients.

	(median 1.77)	(median 1.73)	(median 1.73)	
	Min-Max: 0.00–5.01	Min-Max: 0.00–4.67	Min-Max: 0.4–4.67	

**Parameters**	**At Risk (** * **n** * **= 62)**	**Malnourished (** * **n** * **= 13)**	**Healthy Control ** **(** * **n** * **= 75)**	*** ** * **p** * **-Value at Risk vs. Malnourished**
Age (yr.)	46.35 ± 12.63	38.69 ± 10.26	51.2 ± 13.2 **	*p* = 0.04
	(median 46.35)	(median 38.69)	(median 61)	
	Min-Max: 21.09–71.61	Min-Max: 18.17–59.21	Min-Max: 35–70	
Body weight (kg)	79.38 ± 15.91	55.45 ± 8.71	74.83 ± 14.70 **	*p* = 0.000001
	(median 79.38)	(median 55.45)	(median: 73.60)	
	Min-Max: 47.56–111.20	Min-Max: 38.03–72.87	Min-Max: 49.40–120.00	
Fat mass (kg)	27.29 ± 9.39	13.53 ± 6.10	25.72 ± 9.32 **	*p* = 0.000003
	(median 27.29)	(median 13.53)	(median: 24.83)	
	Min-Max: 8.51–46.07	Min-Max: 1.33–25.73	Min-Max: 7.60–44.00	
Muscle Mass (kg)	28.87 ± 7.31	22.79 ± 3.94	28.90 ± 6.90 **	*p* = 0.005
	(median 28.87)	(median 22.79)	(median: 28.90)	
	Min-Max: 14.25–43.49	Min-Max: 14.91–30.67	Min-Max: 15.00–47.00	
Body fat (%)	34.23 ± 8.07	24.68 ± 7.34	35.4 ± 7.4 **	*p* = 0.0002
	(median 34.23)	(median 24.68)	(median 33)	
	Min-Max: 18.09–50.37	Min-Max: 10.00–39.36	Min-Max: 13–50	
Total water (L)	38.07 ± 8.88	30.75 ± 4.85	35.4 ± 7.4	*p* = 0.005
	(median 38.07)	(median 30.75)	(median 33.7)	
	Min-Max: 20.31–55.83	Min-Max: 21.05–40.45	Min-Max: 14.5–52.4	
BMI (kg/m^2^)	27.52 ± 4.74	19.53 ± 1.84	27.35 ± 4.35 **	*p* = 0.000
	(median 27.52)	(median 19.53)	(median: 27.10)	
	Min-Max: 18.04–37.00	Min-Max: 15.85–23.21	Min-Max: 20.30–37.80	
Body weight loss (kg)	0.00 ± 0.00	1.75 ± 2.53	0 **	*p* = 0.000
	(median 0.00)	(median 1.75)	(median 0)	
	Min-Max: 0.00–0.00	Min-Max: 3.31–6.81	Min-Max: 0	
Total cholesterol (mg/dl)	224.37 ± 47.40	194.77 ± 46.64	198.3 ± 42.3 **	*p* = 0.045
	(median 224.37)	(median 194.77)	(median: 198.3)	
	Min-Max: 129.57–319.17	Min-Max: 101.49–288.05	Min-Max: 130.0–290.0	
HDL cholesterol (mg/dl)	61.10 ± 16.76	66.98 ± 20.99	57.9 ± 14.3	*p* = 0.278
	(median 61.10)	(median 66.98)	(median 68)	
	Min-Max: 27.58–94.62	Min-Max: 25.00–108.96	Min-Max: 22.8–87	
LDL cholesterol (mg/dl)	137.43 ± 37.82	110.92 ± 34.04	68.1 ± 27.1 **	*p* = 0.026
	(median 137.43)	(median 110.92)	(median 66.5)	
	Min-Max: 61.79–213.07	Min-Max: 42.84–179	Min-Max: 23–177	
hsCRP (mg/L)	2.88 ± 2.20	4.96 ± 3.20	2.1 ± 7.2 **	*p* = 0.006
	(median 2.88)	(median 4.96)	(median: 2.1)	
	Min-Max: 0.21–7.28	Min-Max: 0.20–11.36	Min-Max: 0.2–20.0	
Irisin (ng/mL)	237.14 ± 233.67	75.65 ± 85.28	765.85 ± 799.35 **	*p* = 0.020
	(median 237.14)	(median 75.65)	(median: 369.39)	
	Min-Max: 15.11–704.48	Min-Max: 15.40–246.21	Min-Max: 15.9–2704.4	
IL-6 (pg/mL)	1.97 ± 3.38	1.32 ± 1.22	1.32 ± 1.22	*p* = 0.494
	(median 1.97)	(median 1.32)	(median: 1.32)	
	Min-Max: 0.00–8.73	Min-Max: 0.00–3.76	Min-Max: 0.30–3.76	
BDNF (pg/mL)	387.47 ± 647.82	365.77 ± 436.21	365.77 ± 436.21	*p* = 0.909
	(median 387.47)	(median 365.77)	(median: 365.77)	
	Min-Max: 0.00–1683.11	Min-Max: 0.00–1238.19	Min-Max: 0.41–1238.19	
Myostatin (ng/mL)	1.77 ± 1.62	1.73 ± 1.47	1.73 ± 1.47	*p* = 0.935
	(median 1.77)	(median 1.73)	(median: 1.73)	
	Min-Max: 0.00–5.01	Min-Max: 0.00–4.67	Min-Max: 0.4–4.67	

* *p*-value, Student-t, and Mann–Whitney U tests were used for comparisons between MS patients both at risk and malnourished. ** *p* < 0.05, differences between study groups using ANOVA or Kruskal–Wallis test. Data are expressed as mean ± SD, median, and range.

## Data Availability

Data supporting the findings are available on request from the corresponding author.

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
