# Peer review of "The Association Between Myokines, Inflammation, and Nutritional Status in Patients with Multiple Sclerosis"

_biomolecules, 2025, doi:10.3390/biom15050703_

Round 1
Reviewer 1 Report
Comments and Suggestions for Authors
Here, the authors describe a case-control study, comparing levels of serum myokines (myostatin, irisin, BDNF, IL-6, and markers of inflammation (hsCRP) between 92 MS patients and 72 HC. In addition the authors evaluated the associations between these myokines and inflammatory markers with measures of nutritional status and body composition (bioelectric impedance). Cases had lower irisin and s-albumin, and higher hsCRP. Malnourished cases also had lower irisin and higher hsCRP. Malnutrition status amongst the cases was inversely correlated with irisin and positively with hsCRP. From these results, the authors conclude that inflammation and nutritional status interact in people with MS and that nutritional assessment/diet counselling are indicated for this patient population.
The authors have undertaken a number of measures and statistical comparisons in their case-control study. I’m not entirely clear what the hypotheses and objectives are, however. I’m also a bit concerned about the potential for type-1 error, as the ratio of participants and numbers of statistical tests are apt to result in some false positives. Thus, the authors might consider some multiple comparisons adjustment. While there is some biological plausibility for the analyses undertaken, these need to be better justified and supported.
In addition, the authors draw attention to what differences they did find – irisin, s-albumin, and hsCRP – but seem to gloss over the other myokines which did not differ, either between MS cases and controls or by nutritional status, namely myostatin, BDNF, and IL-6.
Finally, the comparisons of myokines and inflammatory markers are limited to cases vs controls and then to correlations between myokines-hsCRP amongst the MS cases identified as malnourished. I would have expected evaluations of cases vs controls among the subset at risk for malnutrition and the malnourished, rather than just correlations amongst the malnourished cases. In terms of inferring potential mechanisms, if there were causal mechanisms at play between myokines and inflammation that interacts with nutritional status, then the magnitudes and significance of associations seen between cases and controls overall should potentiate as nutritional deficiency status increases (from nourished, to at risk of malnutrition, to malnourished). I would suggest the authors undertake such analyses. These could be done through stratified analyses and/or multiplicative interaction approaches. Please consider adding these analyses and speaking to the statistics resultant. In the absence of this, the interpretations drawn as to the myokine-inflammation-nutritional status relationships in MS are not sufficiently supported by the results presented.
I’m unclear why the authors only measured IL-6. Among the many cytokines relevant to MS and particularly to inflammation, for instance IFN-gamma, or anti-inflammatory cytokines like IL-10 and TGF-beta, it’s unclear in the text why IL-6 gets especial attention.
Also, I’m unclear why IL-6 and BDNF are referred to as myokines. I appreciate that in the present analyses the focus is on elements and their relation to muscle and to body composition. However, while the other factors such as myostatin and irisin might be classified as myokines, IL-6 is a cytokine, while BDNF is, as the authors initially describe it, a neurotrophin. If the authors want to evaluate all these factors classed together for the present analyses, that’s fine, but they should be acknowledged as distinct types of compounds with distinct sets of functions. Their relevance to the present analyses, and the context of muscular function/nutrition should be better characterised.
The NRS 2002 assessment is specified in the Methods as comprising unintentional weight loss, BMI, food intake, clinical condition, and age. These component measures and how they were assessed need to be specified. While BMI is presumably just height and weight, is that self-reported or measured? Food intake is a very broad and complicated measure – how was this done? What defined unintentional weight loss and how was this queried/assessed, and over what time frame and what cutoff defines material weight loss? And what is unintentional? What does clinical condition mean? And what do the authors mean by age? I presume the NRS 2002 has some function or scoring framework that aggregates these components, but this needs to be specified. Finally, the cutoffs of what NRS 2002 indicates risk of malnutrition or malnutrition need to be specified.
Similarly, the GLIM Criteria needs more info as to what it comprises, how these were used to define nutritional status, and what the cutoffs were.
That 69% of the MS cases are in the ‘at risk’ category seems to me likely indicative of an overly broad category. However, in the absence of info as to what the cutoffs are, I can’t be certain.
Abstract:
- Please specify how nutritional status was measured and specify what cutoffs define malnutrition and malnutrition risk.
- Please specify statistical methods used to compare between groups.
- In specifying the proportions of MS patients’ nutritional statuses, please specify controls’ and whether differences were significant.
- Please speak to other myokine differences between MS cases and controls. Irisin and BDNF differed but please speak to the others.
- Please specify measures of association (means or differences) in irisin and hsCRP. As written, cases are stated to have lower irisin and higher hsCRP and malnourished cases also lower irisin and higher hsCRP. However, it’s unclear whether these differences are more pronounced in the malnourished cases, or if the differences in cases overall is merely a function of the differences in the malnourished cases. This should be discernible in the Abstract.
- I’m puzzled by the organisation of the Abstract, which seems to have a little Abstract and then an Abstract with sections. I’m unclear if this is just how Biomolecules has authors organise their abstracts or if this is a duplication by the authors. I might suggest just having one or the other.
- Please maintain healthy control terminology, rather than healthy volunteers.
- The terminology regarding the measures of nutritional status (NRS 2002 and GLIM criteria) is unfamiliar to this reviewer. I might suggest defining such terms, as well as indicating what measures underlay their application.
- Mention of s-albumin in the Results of the Abstract has no precedent in the measures enumerated in the Methods.
- The conclusions in the Abstract do not align with the level of evidence presented. No clinical translation or recommendations should be drawn from a case-control study, and certainly not in the Abstract.
Introduction:
- BDNF is given a good deal of explanation as to its role in the brain. However, it’s unclear why serum BDNF would be relevant, or why this peripheral measure would have bearing on the brain or CNS levels of BDNF. Please justify.
- Please conclude the Introduction by enumerating the objectives of this article.
Methods:
- What would prevent participants from undertaking the bioimpedance measurement? Did they need to be able to stand? Or was this reflecting their refusal to undertake the measurement? Please clarify.
- The biochemical parameters makes mention of lipid and complete blood count measures. These didn’t appear anywhere previous to this, in the Abstract or Introduction. If these measures aren’t to be used, that needs to be justified. They seem quite relevant. For instance, there are inflammation scores one can estimate from blood count fractions, which seems quite relevant to the present study. Lipids are obviously quite germane to nutrition, as well as to inflammation, and also to MS. Why were these measures not utilised? If they aren’t to be used in the present manuscript, I might suggest not mentioning in the article.
Results:
- Controls are specified as having ‘good’ nutritional status. Good is not one of the levels of nutritional status enumerated for either NRS or GLIM. Also, do the authors refer to both NRS and GLIM in making this characterisation?
- Can the authors provide mean (SD) for NRS and GLIM for cases and controls?
Figures/tables:
- Please ensure axes are labelled.
- Ensure all abbreviations are defined in tables/figures, and for those presenting statistical comparisons, please ensure the methods used are specified.
- For Figure 2, are these meant to refer to NRS or GLIM or both combined? I might suggest presenting each individually. I would also suggest presenting statistics for cases and controls, not just cases.
- Table 1 should include all the components comprising the NRS and GLIM nutritional status measures. As presented, only BMI is shown and this has no difference between groups. So what then underlies the differences reported in NRS and GLIM between cases and controls?
- Please maintain consistency of presentation of parameters between tables. Table 1 only has some bioimpedance metrics but Table 2 has others. Table 1 only has total cholesterol while Table 2 also has HDL and LDL.
- I’m unclear what some of the letters in the parameter names in Table 2 mean, for instance the E at the end of the IL-6.
Author Response
Thank you very much for your constructive feedback and thorough review. We profoundly appreciate your time and effort in helping us improve our manuscript. Your feedback is invaluable in enhancing the scientific depth, clarity and overall quality of our work. Below are the answers to all questions.
Comments 1.
The authors have undertaken a number of measures and statistical comparisons in their case-control study. I’m not entirely clear what the hypotheses and objectives are, however. I’m also a bit concerned about the potential for type-1 error, as the ratio of participants and numbers of statistical tests are apt to result in some false positives. Thus, the authors might consider some multiple comparisons adjustment. While there is some biological plausibility for the analyses undertaken, these need to be better justified and supported.
Response 1.
Our hypothesis is that nutritional status in MS correlates with increased levels of circulating myokines and cytokines. The objectives of our study was to assess whether the myokines and cytokine are associated with nutritional status in MS patients. According to Reviewer comments we corrected objectives.
We agree with the comments and applied a post-hoc Bonferroni correction to eliminate type-1 error. The Bonferroni test modifies the significance level to control for type-1 errors, or false positives. The Bonferroni test takes into account the number of comparisons made.
Comments 2.
In addition, the authors draw attention to what differences they did find – irisin, s-albumin, and hsCRP – but seem to gloss over the other myokines which did not differ, either between MS cases and controls or by nutritional status, namely myostatin, BDNF, and IL-6.
Resmonse 2.
In accordance with the Reviewer's comments, we have added information that we did not find statistically significant differences in the levels of myostatin, BDNF, and IL-6 between the studied groups of healthy and also well-nourished and malnourished MS patients.
Comments 3.
Finally, the comparisons of myokines and inflammatory markers are limited to cases vs controls and then to correlations between myokines-hsCRP amongst the MS cases identified as malnourished. I would have expected evaluations of cases vs controls among the subset at risk for malnutrition and the malnourished, rather than just correlations amongst the malnourished cases. In terms of inferring potential mechanisms, if there were causal mechanisms at play between myokines and inflammation that interacts with nutritional status, then the magnitudes and significance of associations seen between cases and controls overall should potentiate as nutritional deficiency status increases (from nourished, to at risk of malnutrition, to malnourished). I would suggest the authors undertake such analyses. These could be done through stratified analyses and/or multiplicative interaction approaches. Please consider adding these analyses and speaking to the statistics resultant. In the absence of this, the interpretations drawn as to the myokine-inflammation-nutritional status relationships in MS are not sufficiently supported by the results presented.
Response 3.
As suggested, we presented the results of a comparison between a control group and well-nourished MS patients and a control group and patients at risk of MS. (Table 2)
Multivariate regression analysis model with age, hs CRP, FFMI, albumin, myokines and GLIM/NRS 2002 does not show statistical significance. Unfortunately, the external statistician who performed the analysis was not available at the specified time and we do not know whether other models would be statistically significant.
We added to section Statistical Analysis:
Comparisons between two group were conducted using the Student’s t-test for normally distributed variables and the Mann-Whitney U test for non-normally distributed variables, differences between groups were calculated using either ANOVA or the Kruskal–Wallis test. Post-hoc analysis by the Bonferroni method was applied.
Comments 4.
I’m unclear why the authors only measured IL-6. Among the many cytokines relevant to MS and particularly to inflammation, for instance IFN-gamma, or anti-inflammatory cytokines like IL-10 and TGF-beta, it’s unclear in the text why IL-6 gets especial attention.
Response 4.
IL-6 is pleiotropic cytokine produced by a wide variety of immune as well as non-immune cells and they play crucial roles in a variety of autoimmune diseases. In addition to their roles in hematopoiesis and maintenance of immune homeostasis, they regulate diverse cellular processes that affect embryonic development, cognition, intermediate metabolism and aging. The authors point to the important role of IL-6 in MS itself and in muscle metabolism. Elevated IL-6 levels in the CSF were found in individuals with progressive MS, and CSF IL-6 showed positive correlations with the Expanded Disability Status Scale, the Multiple Sclerosis Severity Score, and CSF glial fibrillary acidic protein levels. (PMID: 40175894 DOI: 10.1186/s12883-025-04145-0 )5.
We added information in Introduction
Additionally, we had not further financial support to measured the others cytokines.
Commenst 5.
Also, I’m unclear why IL-6 and BDNF are referred to as myokines. I appreciate that in the present analyses the focus is on elements and their relation to muscle and to body composition. However, while the other factors such as myostatin and irisin might be classified as myokines, IL-6 is a cytokine, while BDNF is, as the authors initially describe it, a neurotrophin. If the authors want to evaluate all these factors classed together for the present analyses, that’s fine, but they should be acknowledged as distinct types of compounds with distinct sets of functions. Their relevance to the present analyses, and the context of muscular function/nutrition should be better characterised.
Resnponse 5.
In accordance with the Reviewer's Comments, we corrected information in Introduction.
Comments 6.
The NRS 2002 assessment is specified in the Methods as comprising unintentional weight loss, BMI, food intake, clinical condition, and age. These component measures and how they were assessed need to be specified. While BMI is presumably just height and weight, is that self-reported or measured? Food intake is a very broad and complicated measure – how was this done? What defined unintentional weight loss and how was this queried/assessed, and over what time frame and what cutoff defines material weight loss? And what is unintentional? What does clinical condition mean? And what do the authors mean by age? I presume the NRS 2002 has some function or scoring framework that aggregates these components, but this needs to be specified. Finally, the cutoffs of what NRS 2002 indicates risk of malnutrition or malnutrition need to be specified.
Similarly, the GLIM Criteria needs more info as to what it comprises, how these were used to define nutritional status, and what the cutoffs were.
Resnpone 6.
NRS 2002 and GLIM are established methods for nutritional status assessment ( Validity of nutrition screening tools for risk of malnutrition among hospitalized adult patients: A systematic review and meta-analysis. Cortés-Aguilar R et al. Clin Nutr. 2024 May;43(5):1094-1116. doi: 10.1016/j.clnu.2024.03.008. Epub 2024 Mar 15. PMID: 38582013
and Guidance for assessment of the inflammation etiologic criterion for the GLIM diagnosis of malnutrition: A modified Delphi approach. Cederholm T, et al. Clin Nutr. 2024 May;43(5):1025-1032. doi: 10.1016/j.clnu.2023.11.026. Epub 2023 Nov 29. PMID: 3823818
The NRS 2002 (Nutritional Risk Screening 2002) was used according to the recommendations of ESPEN (The European Society for Clinical Nutrition and Metabolism) as a tool to assess the risk of malnutrition. NRS 2002 includes questions about unintentional weight loss, BMI, food intake and the presence and severity of the disease, which may affect the risk of malnutrition. Based on the sum of points obtained:
Malnutrition was determined: ≥3 points, 1-2 points – risk of malnutrition, 0 points – without risk.
The GLIM (Global Leadership Initiative on Malnutrition) criteria from 2018 yr. are recommended by ESPEN for malnutrition diagnosis (in hospital and chronically ill patients) Diagnostic evaluation: Patients identified as at risk undergo a detailed evaluation, including two types of criteria:
Phenotypic criteria: Unintentional weight loss: Weight loss >5% in the last 6 months or >10% over a period of more than 6 months.
Low BMI: BMI <20 kg/m² for individuals under 70 years of age and <22 kg/m² for individuals over 70 years of age.
Reduced muscle mass: Assessed using methods such as bioelectrical impedance analysis (BIA), dual-energy X-ray absorptiometry (DXA), or muscle circumference measurements.
Etiologic criteria:
Reduced food intake or absorption: Consumption of ≤50% of energy requirements for more than 1 week or any reduction in food intake lasting more than 2 weeks.
Disease burden or inflammation: Presence of acute or chronic inflammation associated with diseases. A diagnosis of malnutrition is made when at least one phenotypic criterion and one etiologic criterion are met.
In this study, we used NRS 2002 and GLIM to assess risk and malnutrition. We have added information on the criteria for diagnosing malnutrition in the Methods section.
Commenst 7.
That 69% of the MS cases are in the ‘at risk’ category seems to me likely indicative of an overly broad category. However, in the absence of info as to what the cutoffs are, I can’t be certain.
Resnponse 7.
Both methods (NRS-2002 and GLIM) aim to classify patients into a risk group, and in the presence of a chronic disease, especially a chronic inflammatory one, almost the majority of people will meet the risk of malnutrition criteria. Estimating the risk of malnutrition is important because any nutritional interventions are effective at an early stage of disease. We have added information on the criteria for diagnosing malnutrition in the Methods section.
Commenst 10 about Abstract:
We corrected Abstract according Reviwer suggestions. Response:
Background: Recent studies indicate that in progressive multiple sclerosis (MS)-an inflammatory and degenerative disease of the central nervous system-the biological pathways associated with these effects remain poorly understood. Changes in body weight, whether presenting as overweight or underweight, as well as alterations in adipose and muscle tissue, together with chronic inflammation, may contribute to the disease and influence its course. Objective: The case-control study aimed to measure inflammatory markers and myokine levels (myostatin, irisin), brain-derived neurotrophic factor [BDNF] and IL-6 in the serum of patients with multiple sclerosis and healthy control and assess whether the myokines and cytokines are associated with nutritional status. Methods: The study included 92 MS patients and 75 healthy volunteers. Nutritional status was assessed using the NRS (Nutritional Risk Screening) 2002 and GLIM (Global Leadership Initiative on Malnutrition) criteria. Risk of malnutrition or malnutrition were diagnosed based on ESPEN recommendation. Body composition analysis was performed using the BIA method with the InBody 120 analyzer. Routine laboratory parameters (albumin , lipidogrm, complete blood count ) were measured. Myostatin, irisin, BDNF, IL-6, and hsCRP were measured using ELISA methods. Statistical analysis was conducted using Statistica 13.0 software. Comparisons between two group were conducted using the Student’s t-test for normally distributed variables and the Mann-Whitney U test for non-normally distributed variables, the differences between groups were calculated using either ANOVA or the Kruskal–Wallis test. Post-hoc analysis by the Bonferroni method was applied. Results: In the MS group, a high risk of malnutrition (69.0%) and malnutrition (14.0%) were observed. A statistically significant correlation was found between malnutrition (GLIM) and s-albumin (R = 0.2; p < 0.05) as well as hsCRP (R = 0.23; p < 0.05). The MS patient group had significantly lower levels of irisin, higher levels of hs-CRP, and lower s-albumin compared to healthy volunteers. Malnourished patients with MS exhibited significantly lower irisin levels, as well as higher hsCRP in comparison to MS patients at risk or well-nourished. The levels of myostatin and BDNF did not differ depending on nutritional status. Irisin correlated with hsCRP (R Spearman = −0.5; p = 0.01). Conclusions: Our findings highlight the interplay between chronic inflammation, nutritional status, and myokines level in multiple scerosis.
Comments 11.
BDNF is given a good deal of explanation as to its role in the brain. However, it’s unclear why serum BDNF would be relevant, or why this peripheral measure would have bearing on the brain or CNS levels of BDNF. Please justify.
Resposne 11.
There is increasing evidence that brain-derived neurotrophic factor (BDNF) is produced in contracting skeletal muscle and is secreted as a myokine that plays an important role in muscle metabolism. Other authors have studied serum BDNF levels in response to exercise, and we believe that this is an interesting factor linking muscle function and nutritional status.
Cooments 12. Methods:
What would prevent participants from undertaking the bioimpedance measurement? Did they need to be able to stand? Or was this reflecting their refusal to undertake the measurement? Please clarify.
Resnponse 12.
The criterion for qualifying for the study was the ability to stand on a scale and perform bioimpedance, people who were unable to do this were not qualified for the study .
We added to the Methods - the exclusion criterion from the study was: lack of consent and the inability to perform body weight and bioimpedance measurement
Commenst 13.
The biochemical parameters makes mention of lipid and complete blood count measures. These didn’t appear anywhere previous to this, in the Abstract or Introduction. If these measures aren’t to be used, that needs to be justified. They seem quite relevant. For instance, there are inflammation scores one can estimate from blood count fractions, which seems quite relevant to the present study. Lipids are obviously quite germane to nutrition, as well as to inflammation, and also to MS. Why were these measures not utilised? If they aren’t to be used in the present manuscript, I might suggest not mentioning in the article.
Response 13.
whole blood count - information was removed because none statistical differences was observed
lipid profile was routinely determined in patients
We have added the information to the Results section
Commenst 14. Results:
- Controls are specified as having ‘good’ nutritional status. Good is not one of the levels of nutritional status enumerated for either NRS or GLIM. Also, do the authors refer to both NRS and GLIM in making this characterisation?
- Can the authors provide mean (SD) for NRS and GLIM for cases and controls?
Response 14.
We added information in Method section
We added in Results section: Patients received an average of 0.58 ± 0.6 points in NRS 2002 and 1.17 ± 0.37 points in GLIM.
Comments 15 Figures/tables:
- Please ensure axes are labelled Response - We improved Figure 2
- Ensure all abbreviations are defined in tables/figures, and for those presenting statistical comparisons, please ensure the methods used are specified - We added information about ststistics
- For Figure 2, are these meant to refer to NRS or GLIM or both combined? I might suggest presenting each individually. I would also suggest presenting statistics for cases and controls, not just cases - We added data on Figure 2.
- Table 1 should include all the components comprising the NRS and GLIM nutritional status measures. As presented, only BMI is shown and this has no difference between groups. So what then underlies the differences reported in NRS and GLIM between cases and controls? -
The control group scored 0 points in both the NRS 2002 and GLIM assessments.
- Please maintain consistency of presentation of parameters between tables. Table 1 only has some bioimpedance metrics but Table 2 has others. Table 1 only has total cholesterol while Table 2 also has HDL and LDL - We added data to the Table
- I’m unclear what some of the letters in the parameter names in Table 2 mean, for instance the E at the end of the IL-6.
We have eliminated unnecessary letters.
Additionally, we added Fig.3 and 4. The differences in irisin and hsCRP levels in MS patients according nutritional status (NRS 2002).
Fig.3 The differences in irisin levels in MS patients according nutritional status (NRS 2002). Kruskal Wallis test (Chi 2= 10.97; df = 3 p = 0.01)
Fig.4 The differences in hsCRP levels in MS patients according nutritional status (NRS 2002). Kruskal Wallis test (Chi 2 kwadrat= 11.88; df = 3 p =0.00)

Reviewer 2 Report
Comments and Suggestions for Authors
In the article "The Association between Myokines, Inflammation, and Nutritional Status in Patients with Multiple Sclerosis" the authors aim to demonstrate that chronic inflammation affects nutritional state in MS patients, exacerbating progression of the disease.
The article is interesting, well-written, and the rationale and aims are clearly stated.
Minor concerns:
1. The authors stated that they did not differentiate between MS phenotypes. However, different MS phenotypes also imply varying degrees of disease activity and baseline inflammation status. While I understand that standardizing the samples is quite difficult, would it be possible instead to verify whether the different nutritional states (well-nourished, at risk, malnourished) correlate with relapsing or remitting phases? If malnutrition is linked to inflammation status, I wonder whether the 16% of well-nourished MS patients were those in the remitting phase. This correlation could further support the hypothesis.
2. I notice that Figure 1 lacks a representation of the control group, which I think should be included for better comparison.
Author Response
Comments 1.
"The Association between Myokines, Inflammation, and Nutritional Status in Patients with Multiple Sclerosis" the authors aim to demonstrate that chronic inflammation affects nutritional state in MS patients, exacerbating progression of the disease. The article is interesting, well-written, and the rationale and aims are clearly stated. Minor concerns: 1. The authors stated that they did not differentiate between MS phenotypes. However, different MS phenotypes also imply varying degrees of disease activity and baseline inflammation status. While I understand that standardizing the samples is quite difficult, would it be possible instead to verify whether the different nutritional states (well-nourished, at risk, malnourished) correlate with relapsing or remitting phases? If malnutrition is linked to inflammation status, I wonder whether the 16% of well-nourished MS patients were those in the remitting phase. This correlation could further support the hypothesis.
Response 1.
Thank you for your valuable Review, below we respond to all comments.
We agree that different forms of the disease, as well as the presence of relapse or remission, may significantly affect both inflammatory status and nutritional condition.
In our study, we did not include these variables, which we recognize as a limitation. This was primarily due to the cross-sectional nature of the project and the clinical diversity of the group — patients were at various stages of the disease and undergoing different treatment regimens. Unfortunately, since the data collection was completed some time ago, we no longer have the possibility to retrieve additional information or follow up with patients. Access to complete medical records is also limited, which prevents us from performing the retrospective analysis suggested.
Comments 2.
I notice that Figure 1 lacks a representation of the control group, which I think should be included for better comparison.
Response 2.
In Figure we presented result of nutritional status assessment in MS patients, Control group presented good nutritional status in all cases. We added data on the Figure.
We apologize for mistake in numbering of figures.
